# In Vitro Effect of Three-Antibiotic Combinations plus Potential Antibiofilm Agents against Biofilm-Producing *Mycobacterium avium* and *Mycobacterium intracellulare* Clinical Isolates

**DOI:** 10.3390/antibiotics12091409

**Published:** 2023-09-06

**Authors:** Sara Batista, Mariana Fernandez-Pittol, Lorena San Nicolás, Diego Martínez, Marc Rubio, Montserrat Garrigo, Jordi Vila, Griselda Tudó, Julian González-Martin

**Affiliations:** 1Unitat de Microbiologia, Department de Fonaments Clínics, Facultat de Medicina i Ciències de la Salut, Universitat de Barcelona, c/Casanova 143, 08036 Barcelona, Spain; sbatista@recerca.clinic.cat (S.B.); mjferandez@clinic.cat (M.F.-P.); jvila@clinic.cat (J.V.); 2ISGlobal Barcelona, Institute for Global Health, c/Rosselló 132, 08036 Barcelona, Spain; 3Servei de Microbiologia, CDB, Hospital Clínic de Barcelona, c/Villarroel 170, 08036 Barcelona, Spain; lsannic@clinic.cat (L.S.N.); dmartine@clinic.cat (D.M.); 4Servei de Microbiologia, Fundació de Gestió de l’Hospital de la Santa Creu i Sant Pau, c/Sant Quintí 89, 08026 Barcelona, Spain; mrubiobu@santpau.cat (M.R.); mgarrigo@santpau.cat (M.G.); 5Institut d’Investigació Biomèdica Sant Pau (IIB Sant Pau), c/Sant Quintí, 89, 08026 Barcelona, Spain; 6CIBER of Infectious Diseases (CIBERINFEC), Instituto de Salud Carlos III, 28029 Madrid, Spain

**Keywords:** biofilm, antibiotic combinations, *Mycobacterium avium* complex, antibiotics, antibiofilm agents

## Abstract

Patients with chronic pulmonary diseases infected by *Mycobacterium avium* complex (MAC) often develop complications and suffer from treatment failure due to biofilm formation. There is a lack of correlation between in vitro susceptibility tests and the treatment of clinical isolates producing biofilm. We performed susceptibility tests of 10 different three-drug combinations, including two recommended in the guidelines, in biofilm forms of eight MAC clinical isolates. Biofilm developed in the eight isolates following incubation of the inoculum for 3 weeks. Then, the biofilm was treated with three-drug combinations with and without the addition of potential antibiofilm agents (PAAs). Biofilm bactericidal concentrations (BBCs) were determined using the Vizion lector system. All selected drug combinations showed synergistic activity, reducing BBC values compared to those treated with single drugs, but BBC values remained high enough to treat patients. However, with the addition of PAAs, the BBCs steadily decreased, achieving similar values to the combinations in planktonic forms and showing synergistic activity in all the combinations and in both species. In conclusion, three-drug combinations with PAAs showed synergistic activity in biofilm forms of MAC isolates. Our results suggest the need for clinical studies introducing PAAs combined with antibiotics for the treatment of patients with pulmonary diseases infected by MAC.

## 1. Introduction

Among the mycobacteria genera, the pathogen that causes the most deaths worldwide per year is *Mycobacterium tuberculosis* [1]. However, infections caused by non-tuberculous mycobacteria (NTM) are on the rise, and their relevance will likely be notable in the coming years [2]. The annual prevalence of NTM within all age groups varies from 1.4 to 6.6 cases per 100,000 individuals across four regions of the United States [3]. The prevalence of pulmonary NTM is not consistent across regions, genders, or racial/ethnic groups [4]. The incidence of these infections could have an impact similar to that of tuberculosis in some countries. The most frequently isolated NTM species are those included in the *Mycobacterium avium* complex (MAC), followed by *Mycobacteroides abscessus* and, to a lesser extent, *Mycobacteroides chelonae*, *Mycobacterium kansasii*, *Mycolicibacterium fortuitum*, and *Mycobacterium xenopi* [5]. The *MAC* includes nine species that share similar characteristics, with the most relevant species being *M. avium*, *Mycobacterium intracellulare*, and *Mycobacterium chimaera* [6]. The origin of NTMs is environmental; thus, they can be found in soils, plants, and waters of different origins, such as tap water, swimming pools, and hospital water distribution systems [7]. Exposure to these pathogens is common in the general population, and colonization by these species without the development of disease often occurs. In addition, the development of NTM infection may be associated with the immune status of patients [8]. MAC infections can be classified as pulmonary, disseminated, and lymphadenitis, although currently, the most common are lung infections, which mainly affect patients with chronic obstructive pulmonary disease, bronchiectasis, and cystic fibrosis [9]. Patients with lung disease caused by MAC are at a high risk of death following diagnosis, with a pooled estimate of five-year all-cause mortality of 27% [10].

The management and clinical diagnosis of NTM lung infections in patients with chronic respiratory disease is complex due to the poor response to antibiotic regimens and the tendency of chronicity in these patients. NTMs are characterized by their resistance to many of the most common antimicrobials, as well as many of the available antituberculous antibiotics [5,11,12]. Currently, the recommended treatment for these infections is based on the combination of different antibiotics that may vary depending on the species causing the infection [9]. Lung infections caused by MAC are difficult to treat due to the need for prolonged treatment of 18–24 months and the complicated course with frequent toxic effects caused by the medications [13], leading to treatment interruption and treatment failure. It is difficult to differentiate between colonization and infection in patients with MAC lung infections. Their quality of life is seriously affected, with a poor prognosis, both due to the course of the symptoms and the lack of response to current treatments; thus, in many cases, a cure is not achieved. Reinfections and relapses are common, Ref. [2] and it is necessary to follow these patients for years, with a consequent impact on the health system, which requires continuous efforts and funding. There is an urgent need for more effective management of patients with MAC infections.

The chronicity of MAC infections in patients with bronchiectasis and other chronic functional disorders of the lung favors the production of biofilm. Studies on susceptibility to antimicrobial agents do not always show a good correlation with the clinical efficacy of antibiotics [12,14,15,16]. The treatment of patients with MAC infection in whom biofilm has developed is even more challenging. This phenomenon may occur because pulmonary drug concentrations must be much higher than serum concentrations. Drug concentrations should be higher in the lung due to the presence of secretions related to obstructive disease and the production of biofilm by mycobacteria [17], which hinders the access of the antibiotic, and the actual minimum inhibitory concentration (MIC) at the site of infection may be much higher than that detected in vitro. In biofilm-forming NTM, neither the activity of the recommended and new combinations nor the value of the most optimal MIC to treat them are known. There is scarce information about the ability of NTM to produce biofilm or its role in the persistence of infection and lack of response to treatment, although there are data suggesting that biofilm may play a relevant role [17]. The ability to form biofilm is a property of all NTMs, but the intensity of the biofilm depends on the growth conditions, such as the nutrients present in the culture medium [18]. Biofilm formation improves the survival of bacteria by protecting them from environmental stress, such as antimicrobial agents and disinfectants [19,20].

Some studies of rapid-growth NTM have evaluated the activity of combinations of antibiotics and the use of agents with potential antibiofilm action, including detergents such as N-Acetyl-L-cysteine (NAC) or polysorbate 80, achieving synergism [17]. Ibuprofen (IBU) has been shown to have potential to inhibit biofilm development and *quorum sensing* (QS) in *Pseudomonas aeruginosa* [21]. Another research group [22] reported that animals treated with IBU showed statistically significant decreases in the size and number of lung lesions, reductions in bacillary load, and improvements in survival. As a result, IBU can be administered as an adjuvant to tuberculosis treatment. Diallyl disulfide (DDS) has also been shown to inhibit QS genes and virulence factors in *P. aeruginosa* [23]. Similarly, it has been reported that aspirin (ASA) has an inhibitory effect on biofilm formation [24].

Recent reports have described several components of natural origin from numerous fruits, vegetables, and plants with antimicrobial and antibiofilm activity in *Helicobacter Pylori*, *Staphylococcus aureus*, *P. aeruginosa*, *Listeria monocytogenes*, and *Escherichia coli*. Emulsions and essential oils of peppermint and those containing composite eugenol essential oil [25], such as clove [26], oregano, cinnamon, chestnut, and sage—gallic-acid-rich food plants [27]—have been demonstrated to exhibit antibiofilm activity. Different combinations of two antibiofilm agents have also been demonstrated to exhibit antibiofilm activity in *P. aeruginosa* [28,29]. Similarly, ellagic acid, which is found in numerous fruits and vegetables, has shown antibiofilm properties in multiple strains of *Streptococcus agalactiae* [30].

This inhibition of biofilm development might be explained by a surfactant effect on lipids and the stability of disulfide bridges in the biofilm matrix, favoring its destruction. Taking this into account, the use of compounds aimed at breaking the stability of the biofilm could complement the action of antimicrobials.

On the other hand, anti-inflammatory drugs have been linked to a reduction in inflammation at the site of infection in animal models of tuberculosis, contributing to better response to treatment, probably by regulating the intensity of the inflammatory response and achieving a better balance between antimicrobial action and tissue damage caused by immune response [22]. It would be of interest to know if this action can be applied to the stability of biofilm. Hence, in the present study, the effect of combinations of three antibiotics plus potential antibiofilm agents (PAAs) against biofilm forms (BFs) of MAC isolates was analyzed.

## 2. Results

The eight selected MAC strains formed biofilm classified as moderate–intense, with a mean optical density (OD) value of 0.86 (standard deviation (SD) ± 0.35). The mean OD biofilm formation in the presence of ASA, DDS, IBU, and NAC was 0.87 (SD ± 0.35), 1.03 (SD ± 0.42), 0.74 (SD ± 0.30), and 0.75 (SD ± 0.32), respectively.

The MIC_50_ and minimum bactericidal concentration (MBC_50_) values of antibiotics alone in planktonic forms against *M. avium* isolates were 0.25 to 8 µg/mL and 4 to 32 µg/mL, respectively. For *M. intracellulare* the MIC50 and MBC_50_ were 0.25 to 16 µg/mL and 0.5 to 32 µg/mL, respectively. Both species were susceptible (S) to clarithromycin, rifabutin, and rifampicin and resistant (R) to ethambutol (Appendix A). *M. avium* was resistant to moxifloxacin, with intermediate (I) resistance to *M. intracellulare*. No breakpoints were established for bedaquiline (BED) and clofazimine in CLSI for the S/I/R classification for MAC isolates [31]. For BFs, the biofilm bactericidal concentration (BBC_50_) values of the antibiotics were all greater than 32 µg/mL: 32 to 2048 µg/mL in *M. avium* and 64 to 2048 µg/mL in *M. intracellulare* (Table 1). BED and rifabutin (RB) were the most active antibiotics in both species, with MIC and MBC values of 0.5 and 4 µg/mL, respectively, in planktonic forms. In BFs, BBC values were 128 µg/mL in *M. avium* and 64 µg/mL in *M. intracellulare*. Ethambutol (EMB) showed the worst activity against MAC isolates (Table 1).

DMSO exhibited antimicrobial activity against all the MAC isolates tested at concentrations ≥25%. Ethanol exhibited antimicrobial activity against seven out of eight isolates at a concentration of 35% and the remaining at a concentration of 17.5%.

MIC_50_ values are shown in Table 2 with their respective fractional inhibitory concentration index (FICI) values and the percentage of isolates showing synergistic activity of the three-antibiotic combinations alone in planktonic forms for both MAC species. MIC_50_ ranged from 0.06 to 0.25 µg/mL in *M. avium* and from 0.125 to 2 µg/mL in *M. intracellulare* in combinations without PAAs. The combinations including PAAs showed the same MIC values. MBC_50_ values ranged from 0.06 to 0.5 µg/mL and from 0.25 to 4 µg/mL for *M. avium* and *M. intracellulare,* respectively.

Combinations without PAAs in planktonic forms exhibited synergistic activity in 9 of the 10 combinations in at least one tested MAC isolate. The percentage of isolates showing synergistic activity with each antibiotic combination is shown in Table 2. All the *M. intracellulare* isolates showed synergistic activity for the combinations of CLA + BED + CLO, CLA + BED + EMB, MOX + BED + CLO, and MOX + BED + EMB.

BBC_50_ values of the three-antibiotic combinations ranged between 32 and 256 µg/mL in *M. intracellulare* and between 16 and 256 µg/mL in *M. avium*. All 10 combinations showed synergistic activity in *M. intracellulare*, whereas half of the three-antibiotic combinations showed synergistic activity in *M. avium* (Table 3). The synergistic effect increased by more than threefold for all the three-antibiotic combinations when the PAAs were added. This phenomenon was observed in both *M. intracellulare* and *M. avium* isolates. BBC_50_ values ranged from <0.25 to 16 µg/mL and from 1 to 32 µg/mL in *M. intracellulare* and *M. avium*, respectively (Table 4 and Table 5).

## 3. Discussion

This study set out to assess the importance of three-antibiotic combinations and the role of PAAs against BFs in MAC clinical isolates from patients with chronic pulmonary diseases. The main finding was that with the addition of PAAs, the tested three-drug combinations showed synergistic activity in MAC clinical isolates, with BBC values close to those observed in planktonic forms (MBC).

We found that the BBC_50_ values of the seven antibiotics alone tested in BF were between four and six times higher in dilution than in planktonic forms. In agreement with Brown-Elliott et al. [32] and Fröberg et al. [33], we noted that BED was the antibiotic showing the lowest MBC and BBC values against the tested MAC clinical isolates [32], along with rifamycin antibiotics RB and RIF [33]. Therefore, individual antibiotics showed no activity against BF. The tolerance of BFs to antibiotics is well known and has been previously reported in the literature [17,34]. Biofilm acts as a physical barrier against antibiotics, resulting in a lack of correlation between in vitro susceptibility tests and treatment outcomes.

The currently recommended treatment for MAC infections is a combination of three antibiotics based on CLA + RIF + EMB or MOX + RIF + EMB. In cavitated and severe forms, amikacin is added. Therefore, in the present study, 10 different three-antibiotic combinations of the seven selected antibiotics in planktonic forms (including the two combinations mentioned above) were tested (Table 2). The MIC_50_ and MBC_50_ values were low, showing synergistic activity in the 10 combinations in at least 25% of the isolates in both *M. intracellulare* and *M. avium* species, as shown in Table 2. The percentage of synergistic activity was higher in *M. intracellulare* than in *M. avium*. In our area, the susceptibility of *M. intracellulare* to antimycobacterial antibiotics has been reported to be higher than that of *M. avium* [35]. These results clearly demonstrate that MIC values showed a greater reduction in association with the three-antibiotic combinations compared to the use of antibiotics alone. In planktonic forms, the addition of PAAs to the three-antibiotic combinations showed no effect, with MIC values remaining the same or only varying by one dilution.

The same combinations tested in BFs also presented synergistic activity. Remarkably, in BFs, all the three-antibiotic combinations showed synergistic activity in *M. intracellulare*, and eight combinations showed synergistic activity in *M. avium* (Table 3). BBC_50_ values were between four and six dilutions higher than in planktonic forms. These concentrations cannot be administrated in vivo, as the standard doses administered in the treatment regime would not achieve such concentrations.

The effect of adding PAAs to the three-antibiotic combinations in BFs reduced the BBC_50_ values to concentrations similar to those observed when the combinations are used against planktonic forms. The concentrations can be decreased by four to eightfold in dilutions compared to the BBC_50_ values without the addition of PAAs. The resulting concentrations were much more achievable for oral administration. The four tested PAAs presented slight differences, but all showed the same behavior. This phenomenon was detected in all 10 studied combinations, as well as in both *M. avium* and *M. intracellulare*, but was more notable in *M. intracellulare* isolates, since the achieved BBCs were lower than in *M. avium* isolates.

To our knowledge, this is the first report to describe the efficacy of PAAs studied in *Mycobacterium* spp. using the methodology described in the Material and Methods section [34] The presented results are significant in two major aspects. The first is the synergistic activity achieved by the three-antibiotic combinations in both planktonic and biofilm forms, and the second is the role of PAAs in three-antibiotic combinations in BFs. This is an important issue, considering that the treatment regime is lengthy and that the selected combinations can be administered orally. The mechanism of action of these compounds in the *Mycobacterium* genus was noteworthy. As mentioned in a previous study, mycolic acid at the cell wall makes mycobacteria less accessible to antibiotics [17]. It seems reasonable to think that PAAs play an important role in biofilm by disrupting the biofilm matrix and entering through the mycolic acid cell wall. Likewise, several essential oils and natural-source extracts have been demonstrated to cause loss of cell wall and cell membrane integrity and leakage of intracellular substances [36]. Several studies have suggested the putative antibiofilm activity of the tested PAAs [21,37,38]. The QS cell communication system regulates several cell functions, one of which is biofilm formation [39]. IBU, ASA, and DDS have been reported to exhibit antibiofilm activity by interfering with the QS process in fungi and bacteria by inhibiting the activity of molecules involved in the process of adherence in biofilm, bacterial motility, and chemotaxis [23,40,41,42,43,44,45]. A recent review reported that ASA can increase or decrease outer membrane proteins, efflux pumps, and upregulate antibiotic targets [38]. Additionally, ASAs have been shown to partially or totally revert resistance to colistin in yeast [38]. The mechanism of biofilm inhibition by alkyl gallates has already been described; it involves suppression of the production of extracellular polymeric substances and QS signaling, as well as damage to the microbial cell membrane [27]. Similarly, treatment with DDS abolishes the sensitivity of bacteria to environmental stimuli, causing the cells to be in a state of passivation [21] and decreasing the thickness of the biofilm in a concentration-dependent manner [46]. In our study, the tested PAAs did have an antibiofilm effect on MAC isolates individually, but there was a considerable decrease in BBC values when PAAs were added to the three-antibiotic combinations. Several authors have suggested that IBU or AAS have shown efficacy in preventing biofilm formation [21,23,24]. In addition, Oliveira et al. [47] described that IBU demonstrated moderate efficacy in removing biofilms in *Staphylococcus aureus*. As suggested in these studies, one possible explanation for the present results may be that, similarly to Oliveira’s findings, PAAs might interfere with the QS communication between mycobacterial cells present in the dynamic formation and maintenance of the biofilm matrix, favoring biofilm removal and enhancing the antimicrobial activity of antibiotics.

In addition, our data suggest that three-antibiotic combinations other than those recommended might be effective for the treatment of MAC infections—both in planktonic and biofilm forms.

The presented results, demonstrating the in vitro effectivity of several three-drug combinations, as well as the increase in this effectivity with the addition of potential antibiofilm agents (PAAs), are relevant to the design of future treatment strategies for NTM infections in people with chronic pulmonary disease. There is a growing body of literature evaluating and demonstrating the use of different antibiofilm agents based on essential oils and natural compounds derived from plants that are effective either directly or as coadjuvants against different microorganisms [25,28,29,30]. All these studies, as well as others using the same compound applied to other microorganisms [21,23,24,37,38,47], suggest that antibiofilm agents play a role in the treatment of infections involving biofilm. There is an urgent need to improve the therapeutic options for patients with chronic pulmonary diseases, since responses to current treatments are discouraging [9]. The available information suggests the need for clinical studies and clinical assays including antibiofilm agents together with antibiotic treatment. The fact that most antibiofilm agents are natural compounds or drugs approved for use, such as ASA or IBU, will likely facilitate their use in such studies.

## 4. Materials and Methods

### 4.1. Mycobacterium avium–Intracellulare Complex Isolates

Four *M. avium* and four *M. intracellulare* clinical isolates were selected following the criteria of biofilm-forming capacity under the tested condition and showing antimycobacterial sensitivity or having low MIC values in planktonic forms versus the antibiotics chosen in this study. Isolates used in the present study were obtained from the clinical isolates collection of the Microbiology Department of the Hospital Clinic of Barcelona. The eight clinical isolates were originally obtained from respiratory samples of patients with chronic pulmonary disease. All the isolates were previously tested for the ability to form biofilm.

### 4.2. Antibiotics and Potential Antibiofilm Agents (PAAs)

Five antibiotics used in the treatment of non-tuberculous mycobacteria (NTM) infections were selected according to the British Thoracic Society (BTS) guidelines [9]: clarithromycin (CLA), ethambutol (EMB), moxifloxacin (MOX), rifabutin (RB), and rifampicin (RIF). We also included antibiotics with antituberculosis activity (bedaquiline (BED) and clofazimine (CLO)) that have recently been incorporated as a multidrug-resistant tuberculosis treatment to study whether they may be active against MAC. Four chemical compounds with potential antibiofilm activity were studied: acetyl salicylic acid (ASA), diallyl disulfide (DDS), ibuprofen (IBU), and N-acetyl L-cysteine (NAC).

All the antibiotics, as well as the four PAAs, were purchased from Sigma-Aldrich (St. Louis, MO, USA). CLA, BED, and the antibiofilm agent (IBU) were dissolved in dimethyl sulfoxide (DMSO) (Panreac Applichem, Barcelona, Spain) and diluted with sterile distilled water. MOX, EMB, NAC, and ASA were dissolved in sterile distilled water. RF, CLO, and RB were dissolved in N-N- dimethylacetamide (Sigma-Aldrich, St. Louis, MO, USA) and diluted with sterile distilled water. DDS was dissolved in absolute ethanol (Sigma-Aldrich, St. Louis, MO, USA) and diluted with sterile distilled water. All the antibiotics and PAAs were sterilized using a 0.22 µm filter and stored at 20 ºC until use. The highest concentration of the DMSO solvents and ethanol contained in all the experiments was <2.66%.

For planktonic forms, concentrations ranging from 0.25 to 32 µg/mL were tested for antibiotics alone and from 0.06 to 16 µg/mL for the three-antibiotic combinations and for the combination with PAAs. Concentrations from 16 to 2048 µg/mL were tested for antibiotics alone, and concentrations from 0.25 to 1024 µg/mL were tested for the combinations in BFs. In the case of PAAs, serum concentrations of maximum tolerance in adults were used as references: 200 µg/mL for ASA [48] and 100 µg/mL for NAC [49] and IBU [50]. Since the evidence of toxicity with the use of DDS in humans remains unclear [51], we used a 200 µg/mL, the same concentration as ASA. The study combinations were designed considering the following premises: inclusion of CLA (the most active antibiotic against MAC) or MOX, inclusion of at least two additional antibiotics that act on different mycobacterial targets, and selection of oral administration combinations to enhance accessibility. Thus, 10 different combinations of three drugs were proposed using these antibiotics, including those recommended for empirical treatment in the BTS guidelines (CLA + RIF + EMB and MOX + RIF + EMB).

### 4.3. Antimicrobial Susceptibility Testing in Planktonic Forms

The minimum inhibitory concentration (MIC) and the minimum bactericidal concentration (MBC) values were determined using the microdilution method. For individual antibiotics, the three-antibiotic combinations, and the three-antibiotic combinations plus PAAs, the MIC was determined in 96-well plates (Smartech Biosciences, Barcelona, Spain) by adding 100 µL of Middlebrook 7H9 liquid medium (Becton Dickinson, Sparks, MD, USA) to each well. Then, 100 µL of antibiotic was added to the first well, and twofold serial dilutions ranging from 0.5 or 0.06 µg/mL to 32 or 8 µg/mL were performed. Finally, 100 µL of inoculum at a concentration of 1.5 × 10^5^ colony-forming units (CFU)/mL was added (1/1000 dilution of a 0.5 McFarland, using a nephelometer) (PhoenixSpec, Becton Dickinson). The positive control wells contained 100 µL of Middlebrook 7H9 and 100 µL of inoculum. The negative control wells were also included by adding 200 µL of Middlebrook 7H9. The microplates were incubated at 37 °C for 7 days. After incubation, the plates were read using a Vizion System (Sensititre Vizion Digital MIC Viewing System, Thermo Fisher Scientific, Waltham, MA, USA). The MIC value was interpreted as the lowest antibiotic concentration inhibiting mycobacterial growth. Planktonic bacteria were classified as susceptible or resistant according to CSLI guidelines [31]. The MBC was determined by transferring 20 µL from each well of the MIC plate to a second well plate containing 180 µL of Middlebrook 7H9. These plates were then incubated for 1 week at 37 °C. After incubation, the plates were checked for visual growth using the Vizion System. The MBC was interpreted as the lowest concentration without visual growth, assuming that 99.9% of the bacterial cells were killed compared to growth control. For analysis of susceptibility to antibiotics, MIC_50_ and MBC_50_ were calculated. MIC_50_ and MBC_50_ values were defined as the lowest concentration of the antibiotic at which 50% of the isolates were inhibited.

#### Control Testing of the Antibiotic Solvents DMSO and Ethanol in 8 MAC Isolates

In order to test the antimicrobial effect of the solvents, we determined the MIC values using the microdilution method in 96-well plates by adding 100 µL of Middlebrook 7H9 liquid medium to each well. Then, 100 µL of pure DMSO or ethanol was added to the first well, and twofold serial dilutions ranging from 50% to 0.39% for DMSO and 35% to 0.27% for ethanol were performed. Finally, 100 µL of inoculum at a concentration of 1.5 × 10^5^ CFU/mL was added (1/1000 dilution of a 0.5 McFarland, using a nephelometer). The positive control wells contained 100 µL of Middlebrook 7H9 and 100 µL of inoculum. Negative control wells were also included by adding 200 µL of Middlebrook 7H9. The microplates were incubated at 37 °C for 7 days. After incubation, the plates were read using the Vizion System (SWIN® version 3.3.2.7).

### 4.4. Biofilm Formation in Clinical Isolates

In vitro biofilm was formed as previously described [34]. Briefly, biofilm was formed in 96-well plates (Thermo Fisher Scientific, Waltham, MA, USA). The isolates were grown in Middlebrook 7H9 broth. Then, the mycobacterial cultures were homogenized by agitation and adjusted with Middlebrook 7H9 broth to a concentration of 1 × 10^7^ CFU/mL using a nephelometer. Afterwards, 200 µL/well of inoculum (1 × 10^7^ CFU/mL) was seeded in non-treated polystyrene plates (Thermo Fisher Scientific). Each isolate was seeded in 12 wells of six rows. The plates were incubated for 4 weeks at 42 °C in the case of *M. avium* and at 37 °C for *M. intracellulare* strains. Negative controls containing 200 µL of Middlebrook 7H9 were also included. To minimize evaporation, sterile distilled water was added to the surrounding wells, and the plates were covered with a lid. Each isolate was analyzed in duplicate in the same plate. The biofilm-forming capacity of the isolates was determined with the crystal violet (CV) staining method. After 4 weeks, the biofilm formation of the control plate was quantified. The CV methodology was used to quantify biofilm formation by measuring optical density (OD). First, the supernatant of the plates was discarded, and each well was rinsed once with 200 µL of 1× phosphate-buffered saline (PBS). The plates were dried at 62 °C for 1 h, and the wells were dyed with 200 µL of 1% CV. The plates were incubated at room temperature for 10 min. The CV was then removed from each well, washed once with 200 µL of 1× PBS to remove the excess dye, and dried at 62 °C for 1 h. Finally, 200 µL of 33% acetic acid was added to solubilize and homogenize the biofilm. Then, the reading was performed with a microplate spectrophotometer (BioTek Instruments Inc., Winooski, VT, USA) at 580 nm absorbance (A_580_). The wells containing only Middlebrook 7H9 medium were used as blanks, and their mean A_580_ values were subtracted from the wells containing biofilm (Figure 1A).

The classification of biofilm formation was established according to the OD values as non-forming (<0.2), scarce biofilm (0.21–0.4), moderate biofilm (0.41–0.7), and intense biofilm (>0.71). Strains between moderate and intense biofilm forming were selected to perform the assay.

In planktonic cells, the MIC was referenced if the determination was performed in one week of treatment incubation, and the MBC was referenced if two weeks of incubations were performed. In BFs, we described biofilm bactericidal concentration (BBC) if two weeks of incubation were performed. To compare the results between planktonic and biofilm forms, we used MBC and BBC values, respectively.

### 4.5. Biofilm Treatment Study

The BBC was determined [40].

#### 4.5.1. Antibiotic Alone plus PAAs in Biofilm

To calculate the fractional inhibitory concentration index (FICI) in biofilm in the presence of PAAs, the BBC of antibiotics alone with PAAs was determined. Fixed concentrations of PAAs were used. After 4 weeks of incubation for biofilm formation, the plates were treated with individual antibiotics, individual PAAs, three-antibiotic combinations, and three-antibiotic combinations plus PAAs in 96-well plates (Smartech Biosciences). First, the supernatant of the microplate with formed biofilm was discarded; then, 100 µL of the desired antibiotic was added. For antibiotic combinations, the concentrations ranged from 0.25 to 1024 µg/mL, except for antibiotics alone, which were tested in dilutions ranging from 16 to 2048 µg/mL. For antibiotics alone and the three-antibiotic combinations, 100 µL of PAAs was added at a fixed concentration of 100 µg/mL for NAC and IBU, with 200 µg/mL added for DDS and ASA. The plates were then incubated again for another week at 37 °C. All experiments were conducted in triplicate in different plates and in duplicate in each plate.

#### 4.5.2. Determination of Biofilm Bactericidal Concentration

The supernatant was discarded from the treated biofilm plate; then, 100 µL of sterile distilled water was added to each well. Afterwards, the biofilm was disrupted by mechanical smashing with a pipette. Subculturing was performed by seeding 20 µL of each well in a new well containing 180 µL of Middlebrook 7H9. These plates were then incubated for 1 week at 37 °C. After incubation, the plates were checked for visual growth using the Vizion System (Figure 1B). BBC was interpreted as the lowest concentration without visual growth, assuming that 99.9% of the bacterial cells recovered from a biofilm culture were killed compared to the growth control. BBC has also been used to evaluate the efficacy of antibiotics on biofilm-growing bacteria [52].

#### 4.5.3. Determination of the Fractional Inhibitory Concentration Index

##### In Planktonic Forms

MICs in combination with three antibiotics were determined by crossing the individual MIC and the two concentrations below the MIC for each antibiotic with the corresponding concentration of the other antibiotics. Antibiotic interaction was analyzed using the FICI method, as proposed in a previous study by our group [53]. The fractional inhibitory concentration (FIC) was calculated as a quotient between the MIC of the three-antibiotic combination inhibitory concentration (CIC) and the MIC of each antibiotic using the following equation: FICI = FIC_A_ + FIC_B_ + FIC_C_ = (CIC_A_/MIC_A_) + (CIC_B_/MIC_B_) + (CIC_C_/MIC_C_), where the CIC value is the lowest drug concentration that inhibits bacterial growth when the antibiotic acts in combination, and the MIC value is the lowest drug concentration that inhibits bacterial growth when the antibiotic acts individually. The FICI is the addition of the fractional inhibitory concentration (FIC) of each antibiotic present in the combination.

##### In Biofilm-Forming Forms

The FIC was calculated as a quotient between the BBC of the CIC and the BBC of each antibiotic in the presence of PAAs using the same equation as mentioned in the previous section. The results of the FICI analysis were interpreted according to the following criteria: a decrease in two dilutions under the individual MIC or BBC was interpreted as synergistic with a FICI ≤0.75; indifference was considered for values from 0.75 to 4; and a FICI >4 was considered as antagonistic activity. FICI_50_ values were defined as the values of FICI that included 50% of the tested isolates.

## 5. Conclusions

The results of this study show that the addition of potential antibiofilm agents to three-drug combinations of antimycobacterial antibiotics potentiates their activity against biofilm forms of *Mycobacterium avium* complex, achieving a biofilm bactericidal concentration close to that observed against planktonic forms.

In addition, eight combinations of three antibiotics achieved antimicrobial activity comparable to that of the combinations recommended in the guidelines.

The results of this study suggest the need to undertake clinical studies and clinical assays including antibiofilm agents combined with antibiotics in order to improve the therapeutic options of patients with chronic pulmonary diseases due to infection by *Mycobacterium avium* complex.

## Figures and Tables

**Figure 1 antibiotics-12-01409-f001:**
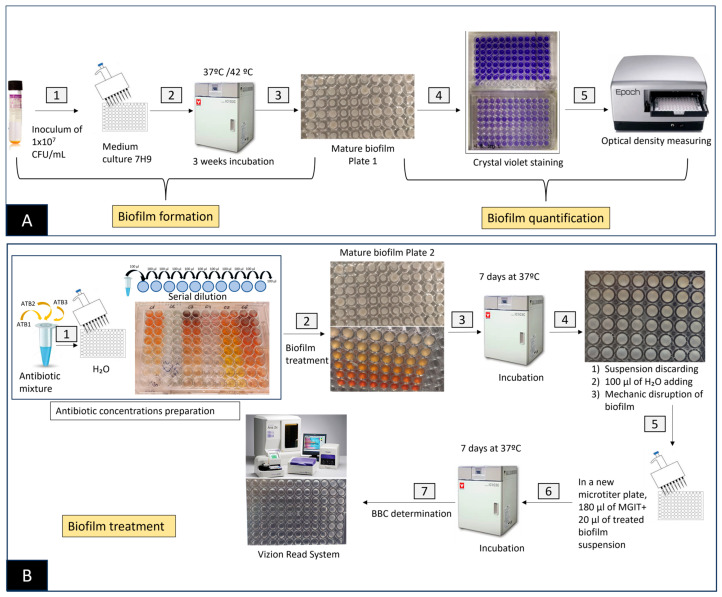
(**A**) Experimental design for biofilm production and quantification by the crystal violet (CV) method: (1) Inoculum preparation; (2) incubation of the 96-well plate containing inoculum; (3) biofilm formation after 3 weeks; (4) crystal violet staining; (5) optical density measurement using a microplate spectrophotometer. (**B**) Experimental design of the antibiotic serial dilution preparation, biofilm treatment with antibiotics, and BBC determination in MAC-producing biofilm isolates: (1) Individual antibiotic/antibiofilm agent mixture and serial dilution preparation using water in a 96-well plate; (2) transfer of prepared antibiotic to the plate with developed biofilm; (3) incubation of the plates with treated biofilm at 37 °C for 1 week; (4) discarding of the suspension of the plates and disruption of the biofilm with a pipette by adding 100 µL of distilled water; (5) transfer of 20 µL of the suspension of the treated biofilm to a microtiter plate containing 180 µL of the fresh medium; (6) incubation of the plates for 1 week at 37 °C; (7) BBC determination by the Vision Read System.

**Table 1 antibiotics-12-01409-t001:** MICs, MBCs, and BBCs of the antibiotics alone in planktonic and biofilm forms in MAC clinical isolates.

	*Planktonic Form*				*Biofilm Form*	
Antibiotic	*M. avium*		*M. intracellulare*		*M. avium*	*M. intracellulare*
	MIC_50_	MBC_50_	MIC_50_	MBC_50_	BBC_50_	BBC_50_
CLA	4	16	2	32	256	256
MOX	4	32	2	4	512	512
BED	0.5	4	0.5	4	128	64
CLO	4	4	16	32	512	256
RB	0.25	4	0.25	0.5	32	512
RIF	0.25	4	0.5	1	256	512
EMB	8	8	2	32	2048	2048

MIC: minimum inhibitory concentration (µg/mL); MBC: minimum bactericidal concentration (µg/mL); BBC: bactericidal biofilm concentration (µg/mL); MAC: *Mycobacterium avium* complex; CLA: clarithromycin; MOX: moxifloxacin; BED: bedaquiline; CLO: clofazimine; RB: rifabutin; RIF: rifampicin; EMB: ethambutol.

**Table 2 antibiotics-12-01409-t002:** MICs, MBCs, and MICs, as well as MBC FICIs, of three-antibiotic combinations in planktonic forms of 4 *Mycobacterium avium* and 4 *Mycobacterium intracellulare* clinical isolates.

	Planktonic Forms											
Combination	*Mycobacterium avium*						*Mycobacterium intracellulare*					
	MIC_50_	MIC_50_ FICI Range	% Isolates Showing Synergistic Activity	MBC_50_	MBC_50_ FICI Range	% Isolates Showing Synergistic Activity	MIC_50_	MIC_50_ FICI Range	% Isolates Showing Synergistic Activity	MBC_50_	MBC_50_ FICI Range	% Isolates Showing Synergistic Activity
CLA + BED + CLO	0.25	0.75–2.5	25	0.25	0.30–4.5	50	0.5	0.31–0.64	100	1	0.19–0.63	100
CLA + BED + EMB	0.25	0.33–1.28	25	0.25	0.17–0.88	100	0.5	0.30–0.50	100	1	0.19–0.41	100
CLA + CLO + EMB	0.25	0.28–1.09	50	0.5	0.13–1.75	50	2	0.17–2.13	50	4	0.38–0.75	100
CLA + RB + EMB	0.06	0.05–1.09	50	0.06	0.03–0.08	100	0.125	0.38–1.5	50	0.25	0.27–0.54	100
CLA + RIF + EMB *	0.125	0.14–1.10	50	0.5	0.22–0.34	100	0.5	0.51–1.5	25	1	1.09–2.3	50
MOX + BED + CLO	0.125	0.39–1.25	50	0.125	0.20–2.13	50	0.25	0.16–0.64	100	1	0.38–0.53	100
MOX + BED + EMB	0.5	0.37–2.56	25	0.5	0.05–0.20	100	0.25	0.15–0.75	100	1	0.19–0.63	100
MOX + CLO + EMB	0.25	0.03–1.09	50	0.5	0.07–0.41	100	2	0.39–2.13	50	4	0.38–1.63	50
MOX + RB + EMB	0.06	0.26–1.28	25	0.06	0.02–0.08	100	0.125	0.25–1.25	50	0.25	0.27–4.75	50
MOX + RIF + EMB *	0.25	0.28–2.03	25	0.5	0.2–0.2	100	0.5	0.57–3	25	1	1.06–2.75	0

MIC: minimum inhibitory concentration (µg/mL); MBC: minimum bactericidal concentration (µg/mL); FICI: fractional inhibitory concentration index; CLA: clarithromycin; MOX: moxifloxacin; BED: bedaquiline; CLO: clofazimine; RB: rifabutin; RIF: rifampicin; EMB: ethambutol; * recommended combinations.

**Table 3 antibiotics-12-01409-t003:** The range of BBCs and FICIs with the three-antibiotic combinations against BFs of MAC clinical isolates (four *M. avium* and four *M. intracellulare*).

Antibiotic Combination	*M. avium*			*M. intracellulare*		
	BBC_50_ (µg/mL)	FICI_50_ Range	% Isolates Showing Synergistic Activity ^1^	BBC_50_ (µg/mL)	FICI_50_ Range	% Isolates Showing Synergistic Activity ^1^
CLA + BED + CLO	64	0.54–>4	25	64	0.31–1.75	50
CLA + BED + EMB	32	0.63–>4	25	64	0.22–1.38	50
CLA + CLO + EMB	256	0.52–>4	50	64	0.13–0.88	50
CLA + RB + EMB	16	1–>4	0	32	0.20–0.44	100
CLA + RF + EMB *	128	0.25–>4	25	64	0.20–0.53	100
MOX + BED + CLO	32	0.22–>4	25	64	0.31–1.63	75
MOX + BED + EMB	32	0.17–>4	50	64	0.22–3.23	50
MOX + CLO + EMB	128	0.43–>4	75	64	0.13–0.44	75
MOX + RB + EMB	64	1–>4	0	256	0.39–2.25	50
MOX + RF + EMB *	32	0.17–>4	25	64	0.25–3	75

BBC: bactericidal biofilm concentration (µg/mL); BF: biofilm form; FICI: fractional inhibitory concentration index; MAC: *mycobacterium avium* complex; ^1^ percentage of isolates showing synergistic activity with FIC values ≤0.75; CLA: clarithromycin; MOX: moxifloxacin; BED: bedaquiline; CLO: clofazimine; RB: rifabutin; RIF: rifampicin; EMB: ethambutol; * recommended combinations.

**Table 4 antibiotics-12-01409-t004:** Range of BBC and FICI_50_ for the three-antibiotic combinations in the presence of PAAs against BFs of *Mycobacterium avium* clinical isolates.

Antibiotic Combination	*M. avium*							
	NAC		IBU		ASA		DDS	
	BBC_50_	FICI_50_ Range	BBC_50_	FICI_50_ Range	BBC_50_	FICI_50_ Range	BBC_50_	FICI_50_ Range
CLA + BED + CLO	2	0.007–0.16 *	4	0.08–0.16 *	2	0.08–0.13 *	4	0.03–0.13 *
CLA + BED + EMB	1	0.005–0.13 *	1	0.005–0.07 *	2	0.01–0.27 *	1	0.005–0.27 *
CLA + CLO + EMB	16	0.05–0.20 *	32	0.08–0.20 *	8	0.03–0.20 *	8	0.05–0.41 *
CLA + RB + EMB	4	0.03–0.26 *	2	0.03–0.16 *	2	0.03–0.13 *	2	0.04–0.34 *
CLA + RIF + EMB **	32	0.03–0.53 *	8	0.03–0.22 *	8	0.07–0.22 *	8	0.03–0.17 *
MOX + BED + CLO	4	0.08–0.14 *	2	0.034–0.11 *	4	0.034–0.11 *	4	0.034–0.08 *
MOX + BED + EMB	4	0.04–0.34 *	4	0.04–0.69 *	4	0.04–0.06 *	4	0.04–0.34 *
MOX + CLO + EMB	16	0.05–0.11 *	8	0.03–0.14 *	8	0.03–0.14 *	8	0.03–0.22 *
MOX + RB + EMB	1	0.06–0.55 *	2	0.06–0.14 *	2	0.07–0.55 *	4	0.04–0.55 *
MOX + RIF + EMB **	16	0.06–0.22 *	8	0.04–0.44 *	8	0.03–0.44 *	8	0.03–0.44 *

BBC: bactericidal biofilm concentration (µg/mL); FICI: fractional inhibitory concentration index; PAA: potential antibiofilm agent; BF: biofilm form; NAC: N-acetyl L-cysteine; IBU: ibuprofen; ASA: acetyl salicylic acid; DDS: diallyl disulfide; CLA: clarithromycin; MOX: moxifloxacin; BED: bedaquiline; CLO: clofazimine; RB: rifabutin; RIF: rifampicin; EMB: ethambutol; * synergistic activity; ** recommended combinations.

**Table 5 antibiotics-12-01409-t005:** Range of BBC and FICI_50_ for the three-antibiotic combinations in the presence of PAAs against BFs of *Mycobacterium intracellulare* clinical isolates.

Antibiotic Combination	*M. intracellulare*							
	NAC		IBU		ASA		DDS	
	BBC_50_	FICI_50_ Range	BBC_50_	FICI_50_ Range	BBC_50_	FICI_50_ Range	BBC_50_	FICI_50_ Range
CLA + BED + CLO	0.5	0.006–0.02 *	<0.25	0.001–0.01 *	<0.25	0.001–0.01 *	0.5	0.006–0.01 *
CLA + BED + EMB	0.5	0.007–0.27 *	<0.25	0.0009–0.01 *	0.5	0.003–0.017 *	0.5	0.007–0.017 *
CLA + CLO + EMB	2	0.007–0.2 *	2	0.007–0.02 *	8	0.008–0.06 *	16	0.008–0.39 *
CLA + RB + EMB	2	0.007–0.027 *	1	0.002–0.22 *	1	0.006–0.22 *	2	0.007–0.01 *
CLA + RIF + EMB **	16	0.05–0.13 *	8	0.01–0.13 *	8	0.01–0.13 *	16	0.03–0.13 *
MOX + BED + CLO	0.5	0.005–0.06 *	<0.25	0.001–0.02 *	<0.25	0.001–0.13 *	<0.25	0.001–0.13 *
MOX + BED + EMB	<0.25	0.0009–0.07*	<0.25	0.009–0.006 *	<0.25	0.0009–0.01 *	0.5	0.002–0.07 *
MOX + CLO + EMB	8	0.04–0.33 *	2	0.008–0.16 *	4	0.008–0.33 *	8	0.02–0.33 *
MOX + RB + EMB	4	0.008–0.05 *	1	0.009–0.01*	4	0.008–0.03 *	2	0.008–0.04 *
MOX + RIF + EMB **	16	0.09–0.13 *	16	0.01–0.1 ^+^	16	0.03–0.1 *	16	0.05–0.19 *

BBC: bactericidal biofilm concentration (µg/mL); FICI: fractional inhibitory concentration index; PAA: potential antibiofilm agent; BF: biofilm form; NAC: N-acetyl L-cysteine; IBU: ibuprofen; ASA: acetyl salicylic acid; DDS: diallyl disulfide; CLA: clarithromycin; MOX: moxifloxacin; BED: bedaquiline; CLO: clofazimine; RB: rifabutin; RIF: rifampicin; EMB: ethambutol; * synergistic activity; ** recommended combinations.

## Data Availability

Data are available within the article.

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
