# Peer review of "In Vitro Effect of Three-Antibiotic Combinations plus Potential Antibiofilm Agents against Biofilm-Producing Mycobacterium avium and Mycobacterium intracellulare Clinical Isolates"

_antibiotics, 2023, doi:10.3390/antibiotics12091409_

Round 1

Reviewer 1 Report

Review Comments

1. Are the isolates either in their planktonic or biofilm forms resistant to any of the conventional drugs? Please include the information on drug resistance or sensitivity of the isolates studied.

2. Please include in your introduction and discussion, recent reports on natural antimicrobial and antibiofilm agents like garlic, gooseberry and clove. This is especially relevant with the global issue of antimicrobial resistance. It is also due to the high concentrations of conventional antibiotics that are required for microbial control.  

For example, Diallyl disulphide used in this study is one of the antimicrobial sulphur compounds found in garlic along with allicin, likewise gooseberry has gallic acid and clove has eugenol and ellagic acid.

3. Some microscopic studies would have leveraged the work. The authors could think of including either fluorescence microscopic or Scanning electron microscopic images to corroborate the antimicrobial and antibiofilm activity of the most potent combination of antibiotics. This will also give the readers an opportunity to see the Mycobacterium biofilm. Even simple bright field microscopic images with acid fast staining of control and antibiotic treatment sets should be fine.

4. Planktonic forming forms and Biofilm forming forms can be written as Planktonic forms and Biofilm forms wherever relevant.

5. Line 73: Grammar: There is a scarce information… Rephrase as:  There is scarce information…

6. Line 249: … similarly to Oliveira’s finding… Rephrase as: … similarly to Oliveira’s findings…

7. Line 253… that other three-antibiotic… Rephrase as: …that the other three-antibiotic…  

Quality of English language is good however minor edits have been suggested.

Reviewer 2 Report

The study introduces new findings about the efficacy of potential antibiofilm agents and their interaction with antibiotic combinations against biofilm-forming MAC isolates. The authors discuss how the agents affect biofilm removal and enhance antimicrobial activity. 

While the abstract provides a concise overview, it could benefit from a slightly clearer presentation of the main findings and their implications for potential treatment strategies.

  The article could further strengthen the introduction by discussing the significance of the MAC infection problem in more depth, providing relevant statistics and emphasizing the need for effective treatment options.

 The authors might consider elaborating on the potential clinical implications of their findings and suggesting avenues for future research, such as in vivo studies or clinical trials.

Recommendation for authors:

 Inclusion of Control Testing: We kindly suggest the authors to incorporate a section on control testing in their study. The utilization of DMSO and Ethanol for solubilization is noted; however, a comprehensive exploration of their effects is recommended. Incorporating control experiments will provide a more thorough understanding of the variables at play and enhance the robustness of the findings.

 Abbreviation Clarification at the Beginning: It is advised that the authors include a clear and concise list of abbreviations and their corresponding explanations at the outset of the article. This approach will aid readers in comprehending the abbreviations encountered early in the text and prevent any confusion that may arise from encountering explanations only later in the article.

 Inclusion of Visuals and Schemes: To enhance the methodology's clarity, we recommend the inclusion of representative images and schematic diagrams. Visual aids can effectively elucidate complex procedures, thereby contributing to the readers' better grasp of the techniques employed.

 These suggestions aim to improve the overall readability, comprehensibility, and visual support of the article while minimizing potential confusion caused by excessive or unexplained abbreviations.

Overall quality of English is ok. We suggest the authors consider a more sparing employment of abbreviations throughout the article. This adjustment is aimed at facilitating smoother reading and ensuring the article's accessibility to a wider audience.

Reviewer 3 Report

Dear authors,

As you mentioned, 8 isolates of MAC were included in this study. Why were the results of MICs, MBCs, and BBCs of single drug used, drug combination, and drug combination with PAAs only shown on two representative isolates instead of 8 isolates? The name of the tested isolates must be provided. Or they were the average/mean value among 4 isolates of each strain? Or you mixed 4 isolates as a culture mixture for testing? This concern must be clarified to the readers when published.

I think there are more results behind this work. The overall results included here are not a good presentation. The manuscript should be straightforwardly reported and need more improvement.

Good Luck!

Kindly check some words and grammar errors.

Round 2

Reviewer 2 Report

Dear Authors,

Thank you for modifying the article based on the review history.

Fine for publishing.

Regards

Reviewer 3 Report

The current version can be accepted. I have no concerns.

Kindly check some words and grammar errors.